# Multimodal MR Image Synthesis via Learning Adaptive Group-wise Interactions

## Abstract

Multimodal MR image synthesis aims to generate missing modality image by fusing and mapping a few available MRI data. Most existing approaches typically adopt an image-to-image translation scheme. However, these methods often suffer from sub-optimal performance due to the spatial misalignment between different modalities while they are typically treated as input channels. Therefore, in this paper, we propose an *Adaptive Group-wise Interaction Network* (**AGI-Net**) that explores both inter-modality and intra-modality relationships for multimodal MR image synthesis. Specifically, groups are first pre-defined along the channel dimension and then we perform an adaptive rolling for the standard convolutional kernel to capture inter-modality spatial correspondences. At the same time, a cross-group attention module is introduced to fuse information across different channel groups, leading to better feature representation. We evaluated the effectiveness of our model on the publicly available IXI and BraTS2023 datasets, where the AGI-Net achieved state-of-the-art performance for multimodal MR image synthesis. *Code will be released*.

## 1 Introduction

Multimodal medical data plays an important role in modern clinical diagnosis and treatment by providing diverse, complementary information about organs and tissues, aiming at enhancing both accuracy and confidence in clinical decision-making. For example, the MRI T1 modality is usually used to indicate human anatomies and the T2 modality can highlight soft tissues. However, factors such as patient non-compliance during scanning, extended scanning times, and the degradation of individual modality hinder the broader adoption of multimodal imaging Thukral (2015); Krupa & Bekiesińska-Figatowska (2015). As a result, it is highly desirable to synthesize missing modalities from a limited number of available multimodal data Iglesias et al. (2013); Huo et al. (2018).

Similar to the image translation task, multimodal medical image synthesis usually requires capturing the interrelationships between modalities, integrating fine-grained features across modalities, and establishing mappings from multiple inputs to a single output. Since multimodal image synthesis leverages information from multiple input modalities, it reduces the complexity of network mapping and enhances the reliability of the synthesis compared to the natural image-to-image translation. However, in clinical scenarios, multimodal images are often misaligned due to factors such as motion artifacts, necessitating alignment through iterative optimization or learnable registration methods, leading to accumulated registration errors.

In recent years, significant progress has been made in the field of multimodal MR image synthesis. Approaches such as learning modality-specific representations and latent representations of multimodal images Zhou et al. (2020); Joyce et al. (2017); Chartsias et al. (2017); Meng et al. (2024) have been extensively studied. Fusion strategies like Multi-Scale Gate Mergence Zhan et al. (2021), attention-based fusion Li et al. (2024), and Confidence-Guided Aggregation Peng et al. (2021) further have been explored. These methods typically involve multiple networks or branches for fine-grained intra-modality feature extraction and use shared fusion networks to capture inter-modality relationships, significantly increasing the training and inference costs of the models. Moreover, they do not address the issue of spatial misalignment in the feature fusion process between different modalities. However, designing an single effective network capable of performing fine-grained intra-

modality feature extraction while capturing inter-modality relationships under spatial misaligned conditions remains an unresolved challenge.

It is widely recognized that the quality of features extracted by neural networks is crucial for medical image synthesis. Deep models are required to extract fine-grained features from different modalities and capture local spatial interactions. Since multimodal images are almost impossible to align perfectly in spatial locations (e.g., different tissue structures in T1 or T2 modality), most traditional convolution designs are challenging to achieve optimal performance.

Therefore, in this paper, we study the task of multimodal MR image synthesis and propose a simple yet effective Adaptive Group-wise Interaction Network (AGI-Net), with the key design of Cross Group Attention and Group-wise Rolling (CAGR) module. Here, Cross Group Attention establishes intra-group and inter-group relationships to suppress inter-modality aliasing noise in the input features, while Group-wise Rolling allows independent adaptive rolling of convolution kernels across groups to adjust the kernel positions for each group (as illustrated in Fig. 1), with the rolling offsets predicted by a *routing function* in a data-dependent manner. These two group-based designs work seamlessly together, effectively capturing inter-modality local spatial relationships under partial misalignment, thus enhancing the network to extract and integrate information across different modalities. The proposed module is a plug-and-play component that can replace any convolution layer. We evaluate our approach on the publicly available IXI [1] and BraTS2023 [2] datasets, and extensive experiments conducted by replacing the convolution module within existing frameworks demonstrate the effectiveness of our AGI-Net, achieving a new state of the art for multimodal MR image synthesis.

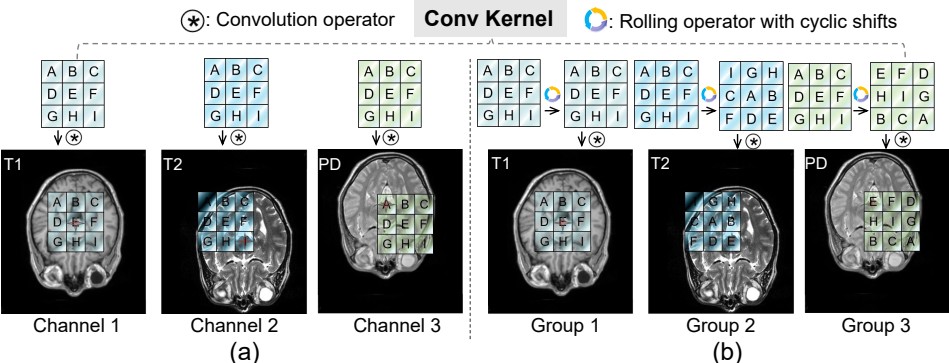

Figure 1: Comparison between rolling convolution and standard convolution with a 3-channel image input. (a) Illustration of standard convolution, where the locations of parameters in each convolution kernel remain fixed across channels. (b) Illustration of rolling convolution, where the convolution weights shift in a data-dependent manner to capture spatial variations across different groups.

## 2 RELATED WORK

**Multimodal Image Synthesis.** Multimodal image synthesis improves upon traditional single-modality image synthesis by extracting intra-modality features and capturing inter-modality correlations, thereby enhancing synthesis accuracy and reliability. Numerous studies have focused on designing various adversarial networks to better capture detailed anatomical structures. For instance, Nie et al. (2018) explores adversarial networks for detailed structure capture, while Dar et al. (2019); Nie et al. (2018); Sharma & Hamarneh (2019) proposes conditional adversarial networks to enhance the synthesis of multi-contrast MR images. Beers et al. (2018) employs progressively grown generative adversarial networks for high-resolution medical image synthesis, and Li et al. (2019) utilizes unified multimodal generative adversarial networks for multimodal MR image syn-

---

[1] https://brain-development.org/ixi-dataset/
[2] https://www.med.upenn.edu/cbica/brats/

thesis. Similarly, Lee et al. (2019) introduces a collaborative generative adversarial network for missing image synthesis.

Another research direction involves developing more adaptive multimodal fusion networks based on extracting modality-specific features with dedicated modality networks. For example, Zhou et al. (2020) uses a hierarchical mixed fusion block to learn correlations between multimodal features, enabling adaptive weighted fusion of features from different modalities. Zhan et al. (2021) employs a multi-scale gate mergence mechanism to automatically learn weights of different modalities, enhancing relevant information while suppressing irrelevant information. Peng et al. (2021) proposes a confidence-guided aggregation module that adaptively aggregates target images in multimodal image synthesis based on corresponding confidence maps. Li et al. (2024) uses a mixed attention fusion module to integrate high-level semantic information and low-level fine-grained features across different layers, adaptively exploiting rich, complementary representative information.

Although the aforementioned approaches have proven effective, they generally focus on constructing various types of adversarial networks and multi-network fusion networks, with limited exploration of the feature extraction challenges posed by imperfect alignment between different modalities.

**Dynamic Convolution.** Standard convolution maintains constant parameters and locations throughout the entire inference process, whereas dynamic convolution allows for flexible adjustments to both parameters and locations based on different inputs, offering advantages in computational efficiency and representational power. Dynamic convolution can typically be classified into two categories: 1) adaptive kernel shape and 2) adaptive kernel parameters. Adaptive kernel shape involves generating suitable kernel shapes according to different inputs. For instance, Dai et al. (2017); Zhu et al. (2019); Wang et al. (2023); Xiong et al. (2024) generates kernel deformations through offsets to capture more accurate semantic information, while Qi et al. (2023) constrains the kernel shape into a snake-like form to capture vascular continuity features.

Adaptive kernel parameters, on the other hand, utilize input-generated kernel weights. For example, Pu et al. (2023); Wang et al. (2024) proposed adaptive rotating kernels to capture objects in various orientations for rotation-invariant object detection. Gao et al. (2019); Kim et al. (2021); Chen et al. (2024) adapts kernel parameters by deformation to handle object deformation while maintaining a consistent receptive field. The method proposed in this paper falls under the category of adaptive kernel parameters. The proposed convolution module employs group-wise rolling kernel parameters, which alleviates the issue of imperfect alignment between modalities and enhances the network's ability to represent multimodal images.

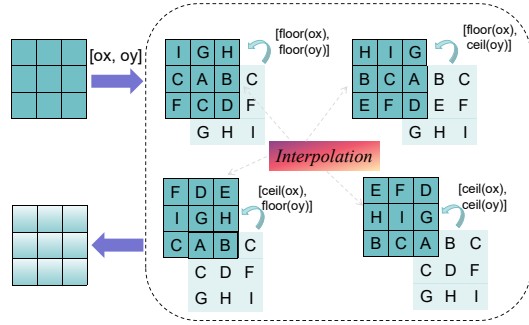
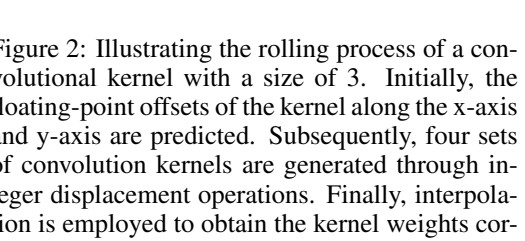
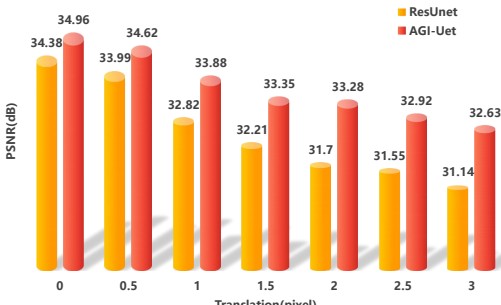

Figure 2: Illustrating the rolling process of a convolutional kernel with a size of 3. Initially, the floating-point offsets of the kernel along the x-axis and y-axis are predicted. Subsequently, four sets of convolution kernels are generated through integer displacement operations. Finally, interpolation is employed to obtain the kernel weights corresponding to the floating-point displacements.

Figure 3: Random translation perturbation test result with the pixel2pixel framework for the (T1, T2)->PD scenario on the IXI dataset.dataset. Random transltion perturbation is applied to the pre-registered T2 modality images in the T1 and T2 pair.

## 3 METHOD

An overview of our core CAGR module is shown in Fig. 4. The CAGR primarily consists of the Cross Group Attention module and Group-wise Rolling module. The function of the Cross Group Attention module is to selectively suppress aliasing noise caused by irrelevant modality features and enhance the expression of relevant modality features by leveraging both intra-group and inter-group information. The Group-wise Rolling module is to dynamically perform group-wise rolling of convolutional kernels based on the predicted offsets. In this section, we begin by introducing the intra-group and inter-group attention mechanisms used in the Cross Group Attention module. Next, we explain the group-wise rolling mechanism for convolutional kernels with specified offsets within the Group-wise Rolling module. Finally, we provide details on the network implementation based on the CAGR module.

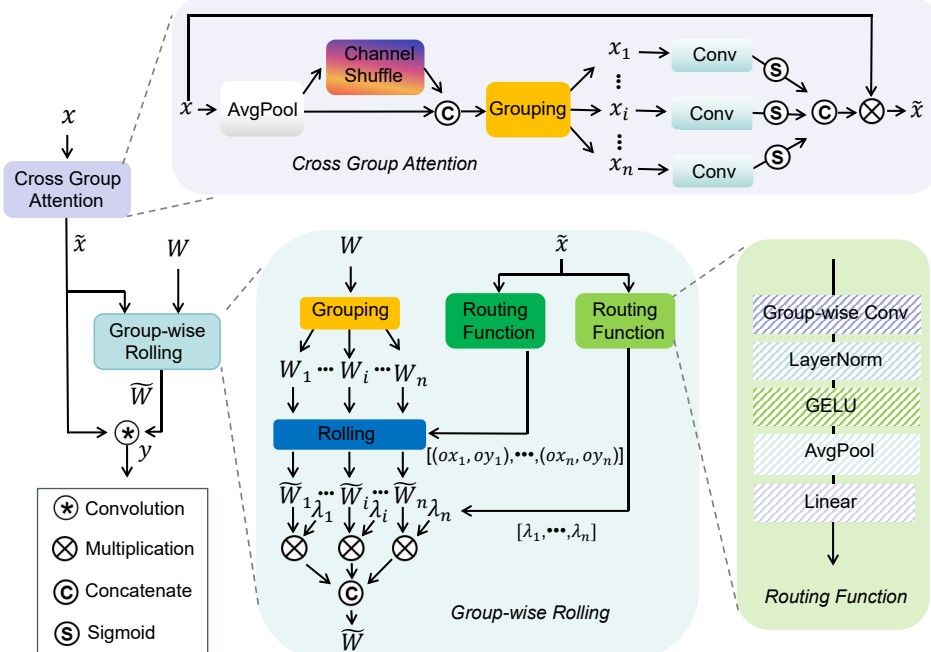

Figure 4: An overview of our proposed CAGR module, which contains two components: Cross Group Attention and Group-wise Rolling. The Cross Group Attention module enhances the input features prior to the Group-wise Rolling module to reduce noise. Following this, the Group-wise Rolling module rolls the convolution kernels in a group-wise manner using the offsets learned from the enhanced input features.

### 3.1 CROSS GROUP ATTENTION

Before performing the rolling convolution operation, for multimodal image synthesis tasks, the input feature $x \in \mathbb{R}^{C_{in} \times H_{in} \times W_{in}}$ can be easily divided into $n$ groups, where each group $x_i, i \in \{1, 2, ..., n\}$ contains feature information related to a specific modality. However, after multiple layers of convolution, each group $x_i$ may incorporate features from other modalities, leading to aliasing noise, which reduces the accuracy of subsequent rolling offset prediction. To address this issue, we employ a Cross Group Attention mechanism. This mechanism selectively suppresses aliasing noise caused by irrelevant modality features and enhances the expression of relevant modality features by leveraging both intra-group and inter-group information.

Specifically, we first apply average pooling to each group feature $x_i$ obtaining the intra-group pool feature $z_i$. Then, using a channel shuffle Zhang et al. (2018); Ma et al. (2018) operation, we rearrange the channels of $z$ to construct inter-group information flow, producing the inter-group feature $z_i^s$. The concatenation of intra-group and inter-group features is then fed through a convolution layer

$F$ followed by a Sigmoid function to generate an attention map.

$$A_i = \sigma(F(concat[z_i, z_i^s])) \tag{1}$$

Finally, the attention map is used to perform element-wise multiplication with the input feature $x$, resulting in enhanced intra-group features and weakened inter-group interference features. The modified feature $\widetilde{x}$ is then used as the input for the subsequent group-wise rolling convolution.

$$\widetilde{x} = \{x_i \odot A_i\}, i \in \{1, 2, ..., n\} \tag{2}$$

## 3.2 Group-wise Rolling

In this section, we provide a detailed explanation of the Group-wise Rolling module within CAGR. This module operates in two main steps: predicting the parameters for rolling convolution based on the enhanced input features by Cross Group Attention, and generating group-wise rolled convolution kernels.

To predict the rolling offsets for each group in a data-dependent manner, we designed a lightweight network called the *routing function*. The *routing function* takes the enhanced image feature $\widetilde{x}$ as input and predicts $n$ groups rolling offsets $[(ox_1, oy_1), \ldots, (ox_n, oy_n)]$ for kernels. Each group independently predicts offsets $\{ox_i\}_{i \in \{1, 2, \cdots, n\}}$ and $\{oy_i\}_{i \in \{1, 2, \cdots, n\}}$ along the x-axis and y-axis, respectively. The overall architecture of the *routing function* is illustrated in Fig. 4. Initially, the enhanced input image feature $\widetilde{x} \in \mathbb{R}^{C_{in} \times H_{in} \times W_{in}}$ is fed into a lightweight Group convolution Krizhevsky et al. (2012) with a kernel size of $3 \times 3$, followed by Layer Normalization ValizadehAslani & Liang (2024); Vaswani (2017) and GELU Hendrycks & Gimpel (2016) activation. The activated features are then average pooled to form a feature vector of dimension $C_{in}$. This pooled feature vector is passed into two separate branches. The first branch is the rolling offset prediction branch, which predicts offsets along the x-axis and y-axis for each group. No activation function is applied to the predicted offsets, enhancing the expressive power of the rolling convolution. The second branch termed the group scale factor prediction branch, is responsible for predicting the scale factor $\lambda$ for each group. It consists of a linear layer with bias and Sigmoid activation. The weights of both the rolling offset prediction and group scale factor prediction branches in the routing function are initialized to zero, and the bias in the group scale factor prediction branch is initialized to one, ensuring stability at the beginning of the training process. Notably, the number of groups $n$ is significantly smaller than the number of input channels of the convolution $C_{in}$. Consequently, it is straightforward to partition the convolution kernels into $n$ distinct groups across the $C_{in}$ channels, with each group having a unique set of offsets.

The standard convolution takes enhanced input features $\widetilde{x} \in \mathbb{R}^{C_{in} \times H_{in} \times W_{in}}$ and kernel weights $W \in \mathbb{R}^{C_{out} \times C_{in} \times k \times k}$, producing the output feature $y \in \mathbb{R}^{C_{out} \times H_{out} \times W_{out}}$. For the multi-channel enhanced input feature $\widetilde{x}$ in multimodal imaging, convolution is applied uniformly across spatial positions with the same kernel weights $\boldsymbol{w}_m \in \mathbb{R}^{C_{in} \times k \times k}$, $m \in \{1, 2, \cdots, C_{out}\}$ performing $C_{out}$ operations to obtain the output feature $y$ with $C_{out}$ channels. In our approach, we divide the kernel weights $W$ into $n$ groups $\boldsymbol{w}_i \in \mathbb{R}^{C_{out} \times C_{in}/n \times k \times k}$, $i \in \{1, 2, \cdots, n\}$ in $C_{in}$ dimension, where kernels from different groups can independently capture features specific to different modalities through the grouping strategy.

$$\begin{aligned} \boldsymbol{W} &= \{\boldsymbol{w}_m \in \mathbb{R}^{C_{in} \times k \times k}\}, m \in \{1, 2, \cdots, C_{out}\} \\ &= \{\boldsymbol{w}_i \in \mathbb{R}^{C_{out} \times C_{in}/n \times k \times k}\}, i \in \{1, 2, \cdots, n\}. \end{aligned} \tag{3}$$

After predicting the rolling offsets and grouping the kernels along the $C_{in}$ dimension, each kernel within the $i$-th group undergoes rolling based on its corresponding offset, resulting in the rolled kernel group. Additionally, each group of kernels is scaled by a learnable scale factor $\lambda_i$, which represents the relative importance of different groups. The final transformation can be expressed by the following equation:

$$\begin{aligned} \widetilde{\boldsymbol{W}} &= \{\lambda_i \times FloatRoll(\boldsymbol{w}_i, (ox_i, oy_i))\}, i \in \{1, 2, ..., n\} \\ &= \{\lambda_i \times ((1 - f(ox_i)) \times (1 - f(oy_i)) \times RollFunc(\boldsymbol{w}_i, (f(ox_i), f(oy_i))) \\ &\quad + f(ox_i) \times (1 - f(oy_i)) \times RollFunc(\boldsymbol{w}_i, (c(ox_i), f(oy_i))) \\ &\quad + (1 - f(ox_i)) \times f(oy_i) \times RollFunc(\boldsymbol{w}_i, (f(ox_i), c(oy_i))) \\ &\quad + f(ox_i) \times f(oy_i) \times RollFunc(\boldsymbol{w}_i, (c(ox_i), c(oy_i))))\}, i \in \{1, 2, ..., n\} \end{aligned} \tag{4}$$

Table 1: Comparison of experimental results on the IXI dataset with existing methods. The evaluation focuses on multimodal image synthesis across three scenarios: (T2, PD)->T1, (T1, PD)->T2, and (T1, T2)->PD. The *Ours* method integrates AGI-Net with pixel2pixel. Notably, the MAE results are scaled by a factor of 100.

| Scenario | (T2, PD)->T1 | | | (T1, PD)->T2 | | | (T1, T2)->PD | | |
|---|---|---|---|---|---|---|---|---|---|
| Method | PSNR(dB) | SSIM(%) | MAE | PSNR(dB) | SSIM(%) | MAE | PSNR(dB) | SSIM(%) | MAE |
| DDPM | 26.55 | 91.26 | 2.20 | 28.88 | 92.02 | 1.89 | 32.24 | 94.77 | 1.35 |
| IDDPM | 26.06 | 90.24 | 2.44 | 28.71 | 90.28 | 1.95 | 32.47 | 94.35 | 1.28 |
| mmGAN | 28.32 | 92.98 | 1.80 | 30.55 | 92.74 | 1.58 | 33.71 | 95.04 | 1.12 |
| pGAN | 28.69 | 93.86 | 1.73 | 30.62 | 92.05 | 1.63 | 33.74 | 95.33 | 1.10 |
| MedSynth | 28.65 | 93.28 | 1.73 | 31.18 | 94.22 | 1.45 | 34.25 | 95.94 | 1.03 |
| pixel2pixel | 28.77 | 93.96 | 1.70 | 31.20 | 94.14 | 1.44 | 34.38 | 96.34 | 1.01 |
| Ours | **29.23** | **94.38** | **1.59** | **31.71** | **94.33** | **1.37** | **34.96** | **96.62** | **0.97** |

where $f(\cdot)$ and $c(\cdot)$ represents $floor$ and $ceil$ function, respectively. The *RollFunc* denotes an operator based on CUDA that supports batch rolling operations on tensors, whereas the built-in roll function in PyTorch does not support batch rolling.

The detailed process of group-wise rolling for convolution kernels is shown in Fig. 2 and Equation 4. First, the floating-point offsets along the x-axis and y-axis are predicted for each group kernel $\hat{w}_i$. Then, integer displacement operations are applied to generate four sets of convolution kernels. Finally, interpolation is used to compute the group kernel weights $\widetilde{w}_i$ corresponding to the floating-point offsets. So the $\widetilde{W}$ can be concatenated in the $C_{in}$ dimension by all group-rolled kernel weights. Notably, the number of kernel parameters in Group-wise Rolling convolution is the same as that in standard convolution, rather than being reduced to $1/n$ of the standard convolution parameters as in Group convolution.

## 3.3 NETWORK ARCHITECTURE

In terms of network implementation, since our proposed CAGR module can be easily integrated as a plug-and-play component into any network structure with convolutional layers, we built the proposed network architecture, AGI-Net, based on the commonly used ResUnet Zhang et al. (2021). ResUnet consists of three down-sampling stages, three up-sampling stages, and a central body stage. Each stage includes two ResBlocks $(z + F(relu(F(z))))$. We replaced the first convolution in each ResBlock within the three down-sampling stages, the body stage, and the first up-sampling stage with the CAGR module to form the new network architecture, referred to as AGI-Net. Ablation studies on replacing different parts of the network and their impact on performance are discussed in the experimental section.

## 4 EXPERIMENT

### 4.1 EXPERIMENT SETTINGS

**Datasets.** We evaluated the proposed method on two publicly available MRI multimodal benchmark datasets: IXI [3] and BraTS2023 [4] LaBella et al. (2024). From the IXI dataset, we selected 577 patients who had T1, T2, and PD-weighted images. The dataset was randomly split into training, validation, and test sets. The training set comprised 500 patients with a total of 44,935 2D images, the validation set contained 37 patients with 3,330 2D images, and the test set had 40 patients with 3,600 2D images. All images were resized to 256x256. Similarly, from the BraTS2023 dataset, we randomly selected T1, T2, and FLAIR images from 580 patients. This dataset was split into 500 patients for the training set, 40 patients for the validation set, and 40 patients for the test set. In terms of 2D images, the training set contained 40,000 images, the validation set 3,200 images, and the test set 3,200 images, with an image size of 240x240. All MRI modalities were normalized to the [0, 1] range using min-max normalization based on the 99.5th percentile maximum value and a minimum value of 0.

---

[3] https://brain-development.org/ixi-dataset/
[4] https://www.med.upenn.edu/cbica/brats/

Table 2: Comparison of experimental results on the BraTS2023 dataset with existing methods. The evaluation focuses on multimodal image synthesis across three scenarios: (T2, FLAIR)->T1, (T1, FLAIR)->T2, and (T1, T2)->FLAIR. The *Ours* method integrates AGI-Net with pixel2pixel. Notably, the MAE results are scaled by a factor of 100.

| Scenario | (T2, FLAIR)->T1 | | | (T1, FLAIR)->T2 | | | (T1, T2)->FLAIR | | |
|---|---|---|---|---|---|---|---|---|---|
| Method | PSNR(dB) | SSIM(%) | MAE | PSNR(dB) | SSIM(%) | MAE | PSNR(dB) | SSIM(%) | MAE |
| DDPM | 23.72 | 87.35 | 2.79 | 23.14 | 88.37 | 2.50 | 20.14 | 84.97 | 4.18 |
| IDDPM | 23.78 | 88.45 | 2.71 | 22.40 | 87.98 | 2.98 | 21.91 | 83.11 | 3.22 |
| mmGAN | 24.92 | 90.68 | 2.28 | 24.21 | 90.64 | 2.21 | 23.39 | 88.14 | 2.54 |
| pGAN | 25.51 | 90.89 | 2.17 | 24.79 | 91.31 | 2.16 | 23.28 | 88.08 | 2.64 |
| MedSynth | 25.70 | 91.11 | 2.10 | 24.84 | 90.98 | 2.13 | 24.00 | 89.06 | 2.40 |
| pixel2pixel | 25.67 | 91.68 | 2.10 | 24.82 | 91.62 | 2.13 | 24.06 | 89.18 | 2.37 |
| Ours | **26.07** | **92.20** | **1.98** | **25.37** | **92.17** | **1.95** | **24.54** | **89.80** | **2.22** |

**Implementation Details.** For training, we set the total number of iterations to 120k using the Adam optimizer with a learning rate of 1e-4 and a batch size of 16. All experiments were conducted in a uniform environment using 4 NVIDIA Tesla V100 GPUs. We utilized the widely recognized Peak Signal-to-Noise Ratio (PSNR), Structural Similarity Index (SSIM), and Mean Absolute Error (MAE) metrics to evaluate the image synthesis quality.

## 4.2 COMPARISON WITH STATE-OF-THE-ART METHODS

We conducted multimodal image synthesis experiments under three scenarios on both the IXI and BraTS2023 datasets, comparing our method against existing approaches using three metrics: PSNR, SSIM, and MAE. The existing methods are categorized into two types based on their generative framework: 1) Diffusion-based methods, which employ multi-step iterative diffusion and sampling, such as DDPM Ho et al. (2020) and IDDPM Nichol & Dhariwal (2021); and 2) Adversarial-based methods, which utilize a single-step approach grounded in the adversarial game between a generator and a discriminator, such as pGAN Dar et al. (2019), mmGAN Sharma & Hamarneh (2019), MedSynth Nie et al. (2018), and pixel2pixel Isola et al. (2017). The *Ours* method integrates AGI-Net with the highly competitive pixel2pixel Isola et al. (2017). As shown in Tab. 1 and Tab. 2, our approach consistently outperforms the existing methods across all multimodal image synthesis scenarios. As shown in Tab. 6, replacing the network from ResUnet to AGI-Net across different methods leads to a significant performance improvement.

## 4.3 ABLATION STUDY

We first conducted ablation studies on different components of the CAGR module and compared it with existing dynamic convolution modules by replacing CAGR with these alternatives. The results demonstrate that CAGR not only significantly outperforms standard convolution methods but also surpasses existing dynamic convolution modules. We further analyzed the impact of the number of groups $n$ and the effects of replacing CAGR at different stages of the network. Lastly, we introduced random translations to the input multimodal images to increase misalignment between modalities, verifying the effectiveness of our method.

Table 3: Ablation studies on the influence of Group-wise Rolling (GR) and Cross Group Attention (CA) module. The experiments were conducted on the IXI dataset in the (T1, T2)->PD scenario, based on the pixel2pixel framework.

| GR | CA | PSNR(dB) | SSIM(%) |
|---|---|---|---|
| - | - | 34.38 | 96.34 |
| ✓ | - | 34.85$_{(\uparrow 0.47)}$ | 96.49$_{(\uparrow 0.15)}$ |
| ✓ | ✓ | **34.96**$_{(\uparrow 0.58)}$ | **96.62**$_{(\uparrow 0.28)}$ |

Table 4: Ablation studies on the impact of different convolution types. The experiments were conducted on the IXI dataset using the (T1, T2)->PD modality synthesis task with a 3-pixel translation, based on the pixel2pixel framework.

| Method | PSNR(dB) |
|---|---|
| ResUnet Zhang et al. (2021) | 31.14 |
| Deform-ResUnet Wang et al. (2023) | 32.09$_{(\uparrow 0.95)}$ |
| ARC-ResUnet Pu et al. (2023) | 32.16$_{(\uparrow 1.02)}$ |
| AGI-Net | **32.63**$_{(\uparrow 1.49)}$ |

**Cross Group Attention and Group-wise Rolling.** We compared the performance of the proposed CAGR module with standard convolution on the IXI dataset. As shown in Tab. 3, we observed that the multimodal image synthesis performance improved through Group-wise Rolling, resulting in a

0.47 dB increase in PSNR on the pixel2pixel Isola et al. (2017) framework. The performance was further enhanced with the inclusion of Cross Group Attention. The experimental results demonstrate that the proposed CAGR module effectively captures richer spatial correspondences and facilitates cross-modal feature fusion across different modalities.

**Different Dynamic Convolutions.** We further compared the performance of the proposed CAGR module with existing dynamic convolution methods on the IXI dataset. These dynamic convolutions can be categorized into two types: one based on adaptive kernel shapes, such as Deformable Convolution Wang et al. (2023), and the other based on adaptive kernel parameters, such as Adaptive Rotated Convolution (ARC) Pu et al. (2023). Following the same replacement strategy as AGI-Net, we constructed Deform-ResUnet and ARC-ResUnet. As shown in Tab. 4, we found that AGI-Net achieved the greatest improvement in multimodal image synthesis performance, with a PSNR increase of 1.49 dB. The experimental results demonstrate that the proposed CAGR module offers significant advantages over previous dynamic convolution methods in the multimodal image synthesis task.

**Number of groups.** An ablation study was conducted to evaluate the impact of different group numbers within the CAGR module. Intuitively, for the 2-to-1 multimodal image synthesis task, setting $n = 2$ is sufficient to capture the spatial relationships between different modalities. As shown in Tab. 5, although $n = 2$ offers a significant performance improvement over standard convolution, the results indicate that as the number of groups increases from 2 to 8, both parameter count and FLOPs decrease while performance improves. However, with $n = 16$, performance begins to degrade, suggesting that $n = 2$ is insufficient to fully capture the misalignment and interrelationships in the multi-channel feature space. Notably, while increasing the number of groups moderately reduces parameters and FLOPs, the best performance ($n = 8$) shows only a slight increase compared to standard convolution. This is because $n$ primarily affects the parameters and FLOPs of the *routing function* and Cross Group Attention's group convolution, with the Cross Group Attention contributing only a small portion to the overall network.

Table 5: An ablation study on the number of groups $n$ conducted in the (T1, T2)->PD scenario of the IXI dataset.

| Network | $n$ | Params(M) | FLOPs(G) | PSNR | SSIM |
|---|---|---|---|---|---|
| ResUnet | - | 17.04 | 141.91 | 34.38 | 96.34 |
| AGI-Net | 1 | 33.58 | 238.84 | 34.76 | 96.49 |
| AGI-Net | 2 | 25.31 | 190.52 | 34.85 | 96.55 |
| AGI-Net | 4 | 21.19 | 166.36 | 34.93 | 96.60 |
| AGI-Net | 8 | 19.15 | 154.28 | **34.96** | **96.62** |
| AGI-Net | 16 | 18.18 | 148.24 | 34.88 | 96.60 |

Table 6: Experimental results of network replacement in different methods for the (T1, T2)->PD scenario on the IXI dataset.

| Method | Network | PSNR | SSIM |
|---|---|---|---|
| mmGAN | ResUnet | 33.71 | 95.04 |
| | AGI-Net | **34.10** | **95.46** |
| pGAN | ResUnet | 33.74 | 95.33 |
| | AGI-Net | **34.33** | **95.59** |
| pixel2pixel | ResUnet | 34.38 | 96.34 |
| | AGI-Net | **34.96** | **96.62** |

**Random translation perturbations test.** We further evaluated the performance variation of the proposed CAGR module under increasing misalignment between input multimodal images. Specifically, we introduced random translation perturbations to one of the two default-registered modalities. The perturbation range was divided into seven difficulty levels, from 0 to 3 pixels, with increments of 0.5 pixels. As shown in Fig. 3, although the performance of both AGI-Net and ResUnet decreases as the magnitude of the translation perturbations increases, AGI-Net exhibits a more gradual decline compared to ResUnet. This indicates that the advantage of the proposed AGI-Net becomes more obvious as the misalignment range increases.

**Replacement strategy.** We conducted experiments to replace the convolutional layers in all stages of ResUnet with the proposed CAGR module. As shown in Table 2, as the number of replaced stages increases, there is a gradual improvement in terms of parameters, FLOPs, and PSNR compared to the baseline model, reaching peak performance when all five initial stages are replaced. Therefore, we selected the configuration with the first five stages replaced to construct our AGI-Net.

**Visualization.** To better illustrate the synthesis performance of our method, we visualize the error maps and regions of interest (ROIs). The experiments were conducted on the IXI test set using a pixel-to-pixel framework. As demonstrated in Fig. 5, the synthesis results of AGI-Net show greater structural consistency with the ground truth compared to ResUnet, proving the superior adaptability of our method in capturing and fusing misaligned features across multiple modalities.

Table 7: An ablation study on the strategy of replacing standard convolutions in the network architecture for the (T1, T2)->PD scenario of the IXI dataset.

| Down1 | Down2 | Down3 | Body | Up1 | Up2 | Up3 | Params(M) | FLOPs(G) | PSNR(dB) |
|-------|-------|-------|------|-----|-----|-----|-----------|----------|----------|
| - | - | - | - | - | - | - | 17.04 | 141.91 | 34.38 |
| ✓ | - | - | - | - | - | - | 17.06 | 144.46 | 34.62 |
| ✓ | ✓ | - | - | - | - | - | 17.12 | 146.95 | 34.73 |
| ✓ | ✓ | ✓ | - | - | - | - | 17.47 | 149.40 | 34.80 |
| ✓ | ✓ | ✓ | ✓ | - | - | - | 18.81 | 151.83 | 34.93 |
| ✓ | ✓ | ✓ | ✓ | ✓ | - | - | 19.15 | 154.28 | **34.96** |
| ✓ | ✓ | ✓ | ✓ | ✓ | ✓ | - | 19.24 | 156.76 | 34.93 |
| ✓ | ✓ | ✓ | ✓ | ✓ | ✓ | ✓ | 19.26 | 159.31 | 34.92 |

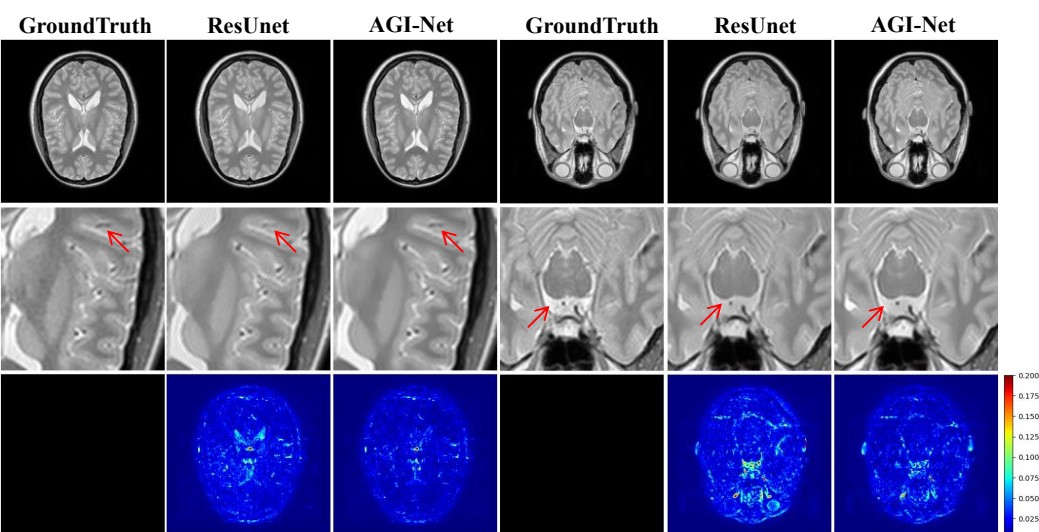

Figure 5: Displays the (T1, T2)->PD synthesis results of pixel2pixel using the IXI dataset. The first row presents the ground truth along with the synthesis results from ResUnet and AGI-Net. The second row shows an enlarged view of the region of interest (ROI), while the third row illustrates the synthesis error map.

### 4.4    LIMITATION AND FUTURE WORK

Although the proposed AGI-Net demonstrates superior performance in multimodal MR image synthesis, addressing potential spatial misalignments between the input multimodal images and the target modality remains challenging. Future work will involve this and focus on developing a unified synthesis framework to further reduce costs in clinical deployments.

## 5    CONCLUSIONS

We present an adaptive group-wise interaction model for multimodal MR image synthesis, featuring two key components: the Cross-Group Attention and Group-wise Rolling modules. The Cross-Group Attention module is designed to fuse both intra-group and inter-group information, effectively mitigating spatial noise between different modality groups. Following this, the convolutional kernels are adaptively rolled in a data-driven manner based on the specific modality groups. This module is flexible and can be integrated into any convolutional backbone for multimodal MR image synthesis. Experimental results show that our AGI-Net significantly enhances image synthesis performance on public multimodal benchmarks while maintaining computational efficiency.

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
