# OpenReview forum: "Multimodal MR Image Synthesis via Learning Adaptive Group-wise Interactions"
_ICLR.cc/2025/Conference — ICLR 2025 Conference Withdrawn Submission_

### Official Review · Reviewer_68VK · 2024-10-28

**Soundness:** 2
**Presentation:** 2
**Contribution:** 2
**Rating:** 3
**Confidence:** 5

**Summary:**

This paper presents the Adaptive Group-wise Interaction Network (AGI-Net) for multimodal MR image synthesis, addressing spatial misalignment issues in existing methods. AGI-Net captures both inter- and intra-modality relationships by applying adaptive rolling on convolutional kernels and incorporating a cross-group attention module for enhanced feature fusion. Tested on the IXI and BraTS2023 datasets, AGI-Net achieves state-of-the-art performance.

**Strengths:**

1.The Adaptive Group-wise Interaction Network (AGI-Net) introduces a novel way to handle multimodal MR image synthesis by capturing both inter- and intra-modality relationships. This dual approach addresses some of the limitations in existing methods that overlook these complex interactions, enhancing feature representation for accurate synthesis.
2. The use of adaptive rolling on convolutional kernels is an innovative technique to capture inter-modality spatial correspondences, tackling challenges in aligning modalities without requiring complex preprocessing.
3. The introduction of a cross-group attention module is a strong addition, enabling effective fusion of information across different channel groups. This approach likely enhances the overall synthesis quality by creating richer feature representations across modalities.

**Weaknesses:**

1. The authors justify their method based on the claimed ineffectiveness of registration techniques for aligning different modalities in a common spatial space. However, given that the study focuses on brain MRIs, which are relatively stationary and free from motion-related issues like blur, standard registration tools, as shown in [1], should be sufficient. This raises questions about the necessity and motivation for developing this new method.
2. While the proposed modules are interesting, multimodal image generation could also be framed as an image translation or style transfer task. Although many GAN-based style transfer approaches have been explored, there are also several diffusion model approaches, such as [2, 3, 4], that could potentially solve this problem without requiring the authors' custom method. A possible alternative would be to learn each modality individually and use one-hot embeddings to specify the desired modality for style transfer.
3. The authors should clarify their experimental setup, specifically what is meant by scenarios like (T1, T2 → FL). Does this imply training a network to generate FL images using T1 and T2 as inputs? Additionally, how would a diffusion model approach support this synthesis setup?
[1]: Iglesias, J.E. A ready-to-use machine learning tool for symmetric multi-modality registration of brain MRI. Sci Rep 13, 6657 (2023). https://doi.org/10.1038/s41598-023-33781-0
[2]: Preechakul K, Chatthee N, Wizadwongsa S, Suwajanakorn S. Diffusion autoencoders: Toward a meaningful and decodable representation. InProceedings of the IEEE/CVF conference on computer vision and pattern recognition 2022 (pp. 10619-10629).
[3]: He, X. et al. (2023). DMCVR: Morphology-Guided Diffusion Model for 3D Cardiac Volume Reconstruction. In: Greenspan, H., et al. Medical Image Computing and Computer Assisted Intervention – MICCAI 2023. MICCAI 2023. Lecture Notes in Computer Science, vol 14226. Springer, Cham. https://doi.org/10.1007/978-3-031-43990-2_13
[4]: Zhang Y, Huang N, Tang F, Huang H, Ma C, Dong W, Xu C. Inversion-based style transfer with diffusion models. InProceedings of the IEEE/CVF conference on computer vision and pattern recognition 2023 (pp. 10146-10156).

**Questions:**

1. Please provide explanation why Brain MRI cannot be corrected with registration method or perform additional experiments on cardiac MRI where it is more difficult for alignment due to the respiratory motion.
2. Please consider additional methods including using condition in diffusion model to guide the style transfer to generate different modality. In this way, no alignment is needed since the same subject with different modality is not learned together.
3. Please elaborate on your experiment settings as addressed previously.

---

### Official Review · Reviewer_tP42 · 2024-11-03

**Soundness:** 2
**Presentation:** 3
**Contribution:** 2
**Rating:** 3
**Confidence:** 4

**Summary:**

This paper proposes an Adaptive Group-wise Interaction Network (AGI-Net) that explores both inter-modality and intra-modality relationships for multimodal MR image synthesis.

**Strengths:**

(1) This paper presents a Cross Group Attention, which establishes intra-group and inter-group relationships to suppress inter-modality aliasing noise in the input features.
(2) This paper presents a Group-wise Rolling strategy that allows independent adaptive rolling of convolution kernels across groups to adjust the kernel positions for each group.
(3) The experimental results on two datasets show the effectiveness of the propsoed model.

**Weaknesses:**

(1) This paper should explicitly delineate the contributions of the proposed MR synthesis method, highlighting its unique strengths and potential impact on the field.

(2) The current comparison methods are inadequate. To substantiate the efficacy of the proposed model, the experimental section should incorporate a broader range of state-of-the-art (SOTA) synthesis methods for a comprehensive evaluation.

(3) The authors utilize a cross-group attention strategy to forge intra-group and inter-group connections, thereby mitigating inter-modality aliasing noise in the input features. However, since cross-attention is a prevalent strategy in related works, it is crucial to elucidate the primary distinctions and superior aspects of this approach compared to existing methods.

**Questions:**

(1) This paper should explicitly delineate the contributions of the proposed MR synthesis method, highlighting its unique strengths and potential impact on the field.

(2) The current comparison methods are inadequate. To substantiate the efficacy of the proposed model, the experimental section should incorporate a broader range of state-of-the-art (SOTA) synthesis methods for a comprehensive evaluation.

(3) The authors utilize a cross-group attention strategy to forge intra-group and inter-group connections, thereby mitigating inter-modality aliasing noise in the input features. However, since cross-attention is a prevalent strategy in related works, it is crucial to elucidate the primary distinctions and superior aspects of this approach compared to existing methods.

(4) The authors claim that the proposed method can reduce spatial noise between different modality groups. It is essential to outline the methodology for validating its effectiveness in diminishing spatial noise, possibly through quantitative metrics or comparative analyses.

(5) The current method accommodates two modalities as inputs to synthesize a third. It would be beneficial to explore the feasibility of expanding this approach to incorporate additional modalities (such as using three modalities) to synthesize a missing one, and to discuss the potential implications and challenges of such an extension.

---

### Official Review · Reviewer_Aohc · 2024-11-05

**Soundness:** 2
**Presentation:** 3
**Contribution:** 2
**Rating:** 3
**Confidence:** 4

**Summary:**

This paper aims to address performance issues in existing multi-modal MRI synthesis methods caused by spatial misalignment between different modalities. The proposed method enhances synthesis by using adaptive rolling of convolutional kernels to capture inter-modality spatial correspondences and a cross-group attention module to fuse information across groups.  Experimental validation is conducted on the two brain MRI datasets: IXI and BraTs.

**Strengths:**

1.
The research problem has certain practical significance, as misalignment often occurs between multi-modal medical images.
2.
The manuscript is well-organized, easy to understand and implement.

**Weaknesses:**

1.
The method lacks innovation, as there are already many similar approaches to solving spatial misalignment between multi-modal images. The proposed use of kernel rolling to overcome misalignment is not very novel.
2.
The experimental setup is not very reasonable.
The IXI and BraTS datasets used for validation are already well-registered, which could not effectively reflect the misalignment problem the method aims to solve (though it still shows improvement in results).

To validate the its effect on spatial misalignment, the authors applied pixel translation to the data. This approach is not very realistic, as in practice, MRI data wouldn't be deliberately shifted after registration just to test a method. It is best to use datasets with real misalignments to validate the method.

The real misalignment is in 3D, but the paper only simulate 2D misalignment.

The comparison methods are rather outdated and do not reflect the state-of-the-art level.

**Questions:**

1.
What would be the effects of applying this method to more modalities (not just in a 2-to-1 synthesis setting), and how would the parameter count and FLOPS change? Besides, is it feasible to extend this to 3D image synthesis scenario?
2.
The results of the two DDPM-based methods show a significant gap compared to other CNN-based methods, which is somewhat counterintuitive. Has any analysis been conducted to understand the reasons for this?

---

### Official Review · Reviewer_6fdW · 2024-11-06

**Soundness:** 3
**Presentation:** 3
**Contribution:** 3
**Rating:** 5
**Confidence:** 4

**Summary:**

The paper introduces the Adaptive Group-wise Interaction Network for multimodal MR image synthesis.

The goal is to generate missing MRI modalities using available data while addressing challenges such as spatial misalignment between modalities.

Due to complex multi-network fusion designs, existing approaches often struggle with alignment issues and high computational costs.

**Strengths:**

x. Adaptive kernels: The group-wise rolling approach is unique in its application to address spatial misalignments effectively.

x. Modular design: The CAGR module's plug-and-play nature makes it versatile for enhancing other convolution-based networks. Combination While adaptive and dynamic convolutions have been explored separately, combining them with a targeted group-wise strategy for cross-modality alignment is unique. The adaptive rolling of convolution kernels based on predicted offsets is an interesting innovation that directly addresses the spatial misalignment problem, a common issue in multimodal image synthesis.

x. Solid results: The paper shows comprehensive results across multiple scenarios and datasets, with consistent improvements demonstrated in the metrics.

**Weaknesses:**

**Insufficient justification of design choices**: There is limited explanation for why specific design decisions (e.g., channel shuffling for CGA or the chosen group-wise approach) were made over alternative methods. Clarifying these choices would help emphasize the methodology's uniqueness.

**Lack of downstream task evaluation**: The paper evaluates image quality using traditional metrics like PSNR and SSIM, but does not assess performance on downstream tasks (e.g., segmentation). Adding segmentation as an evaluation metric would provide a stronger demonstration of the synthesized images' clinical utility.

**Absence of comprehensive ablation studies**:
While ablation studies are provided, they may not comprehensively show how the interaction between CGA and GR modules affects the model’s overall performance. This information would help clarify whether the gains come from their combination or if one module is more impactful than the other.

**Questions:**

**Group-wise rolling (GR) assumptions**:

Does the GR module work effectively for non-linear or complex spatial distortions, or does it primarily handle simple shifts? Would there be limitations if the data had non-uniform deformations?

**Interaction between modules**: Can you provide more insight into the interaction between the CGA and GR modules? Which module contributes more to the model's overall performance, or are they equally essential for handling spatial misalignment?

**Theoretical justification for robustness**: Can you provide more theoretical insights into why AGI-Net, particularly the GR module, is effective at correcting misalignments? How do these justifications hold up when dealing with complex spatial distortions?

**Redundancy in operations**: There appears to be some overlap between the functionalities of the CGA and GR modules in adapting features and handling alignment. Could you elaborate on the distinct roles of each module and why both are necessary?

---

### Note · Authors · 2024-11-16

I have read and agree with the venue's withdrawal policy on behalf of myself and my co-authors.